# Improvements in aerosol layer height retrievals from TROPOMI oxygen A-band measurements by surface albedo fitting in optimal estimation

Martin de Graaf[1], Maarten Sneep[1], Mark ter Linden[1], L. Gijsbert Tilstra[1], David P. Donovan[1], Gerd-Jan van Zadelhoff[1], and J. Pepijn Veefkind[1]

[1]Royal Netherlands Meteorological Institute (KNMI), R&D Satellite Observations, De Bilt, The Netherlands

**Correspondence:** Martin de Graaf (martin.de.graaf@knmi.nl)

**Abstract.** The Aerosol Layer Height (ALH), from the Sentinel-5P/TROPOMI L2__AER_LH product, is based on an optimal estimation (OE) approach, fitting cloud-free measurements to synthetic reflectances in the strongest oxygen absorption band, provided by a neural network trained with high resolution simulated reflectances. The ALH has been continuously improved since its release in 2019, focusing especially on (bright) land surfaces, over which the ALH product showed underestimated aerosol layer heights (biased towards the surface). This paper describes the latest updates of the ALH product, that includes first the introduction of the Directional Lambertian-Equivalent Reflectance (DLER) climatology to improve the surface albedo characterisation over land. Second, the paper describes a further improvement, adding the surface albedo in the feature vector of the OE inversion, using the DLER as prior information. Using this approach, the retrievals over land largely match the retrievals over ocean, which have shown a good comparison with validation data since its release, most notably with CALIOP weighted extinction heights. The albedo is fitted for both land and ocean surfaces, but the implementation is different over land and ocean because of the large range of land surface albedos. Over land, the *a priori* surface albedo values are relaxed so the fitting procedure can incorporate the albedo effects in the retrieval over land. Over ocean, the retrievals are optimised by tuning the *a priori* error settings. The current implementation improves retrievals over land with about 1.5 times more converged results, and decreases land-ocean contrasts in the aerosol layer height retrievals. The average difference between CALIOP weighted extinction height decreased for selected cases from about $-1.9$ km to $-0.9$ km over land and from around $-0.8$ km to $+0.1$ km over ocean. An independent verification with ATLID data from EarthCARE showed consistent behaviour between the new operational data and the test cases.

## 1 Introduction

The vertical distribution of aerosols is an important parameter in remote sensing and climate modelling. For example, the retrieval of $NO_2$ using satellite spectrometers (van Geffen et al., 2022) and the retrieval of aerosol optical depth in the ultraviolet (UV) (Torres et al., 1998) are critically dependent on the height of aerosol layers. Climate models need accurate plume injection heights and aerosol profile information to simulate the transport, and physical and chemical transformation, of aerosol plumes. It is essential for aviation safety in case of volcanic ash plumes (e.g. Kahn et al., 2008). In the subtropics, the climatic effects

of aerosols have been shown to depend strongly on the vertical position of the aerosol plumes relative to clouds. Absorbing
aerosol plumes above clouds absorb radiation (de Graaf et al., 2012; Peers et al., 2015), changing the vertical temperature
distribution, affecting cloud stability and lifetime. Aerosols can decrease cloud droplet number density and evaporate cloud
droplets both at the cloud top (e.g. Diamond et al., 2018) and cloud bottom (e.g. de Graaf et al., 2023). However, in case of
absorbing aerosols this process is counteracted at the top if the temperature inversion is strengthened (Johnson et al., 2004),
while absorbing aerosols in the marine boundary layer tend to decrease the atmospheric stability and invigorate cloud formation
(Yamaguchi et al., 2015).

The aerosol layer height is a simplification of the vertical aerosol extinction profile of the atmospheric column above some
point on Earth, reducing the light extinction by aerosols along the zenith to a total aerosol extinction at some point vertically in
the atmosphere. Vertical aerosol extinction profiles can be characterized by lidars. Ground-based lidar networks provide valu-
able information about global distributions of aerosol extinction profiles, like the Latin American Lidar Network (LALINET),
the Asian dust and aerosol lidar observation network (AN-Net), Micro-Pulse Lidar Network (MPLNET) and the European
Aerosol Research Lidar Network (EARLINET), among others, coordinated by the World Meteorological Organization under
the Global Atmospheric Watch (GAW) Aerosol Lidar Observation Network (GALION) (WMO, 2024). These are essential for
the validation of space-based retrievals. For the global characterisation of the vertical aerosol extinction profile from space, the
Cloud-Aerosol LIdar with Orthogonal Polarization (CALIOP) (Winker et al., 2007) onboard Cloud-Aerosol Lidar and Infrared
Pathfinder Satellite Observation (CALIPSO) has been instrumental, providing profiles of light backscatter and extinction at
532 and 1064 nm for seventeen years, from its launch on April 28, 2006 to its scientific mission end on Aug. 1, 2023. Lidars
are active instruments, sending, in its simplest form, an intense pulsed light signal and receiving its backscattered intensity.
While this enables the profiling of the atmosphere vertically, the spatial coverage is necessarily limited to the extent of the light
beam. Passive, polar-orbiting satellite instruments, using backscattered sunlight during the sunlit part of the day, can cover the
Earth almost entirely in one day. However, this comes at the expense of a reduction of the information content, necessitating
the reduction of the aerosol extinction profile to an aerosol layer altitude.

Several different techniques are available for passive instruments, depending on the specific capabilities of the instruments,
described in detail in Xu et al. (2018). The most important ones are stereo photogrammatery for multiangle-viewing instruments
(e.g. Kahn et al., 2007), polarization in the UV for polarimeters (e.g. Dubovik et al., 2011; Wu et al., 2016), and oxygen
absorption spectroscopy by instruments resolving oxygen absorption bands, like the A-band (755–775 nm) and B-band (685–
695 nm). In oxygen absorption spectroscopy, the penetration depth of the scattered light can be quantified by the depth of the
absorption lines: the longer the path through the atmosphere, the deeper the oxygen absorption lines in the absorption bands.
Since oxygen is well mixed and the concentration is accurately known, this can be related to the height of a scattering layer
in the atmosphere (e.g. Pflug and Ruppert, 1993; Kylling et al., 2018). It has recently been applied to measurements in the $O_2$
A-band from instruments like Earth Polychromatic Imaging Camera (EPIC) (Xu et al., 2019), Orbiting Carbon Observatory-2
(OCO-2) (Zeng et al., 2020), and Ocean and Land Colour Imager (OLCI) (Jänicke et al., 2023).

The different techniques and instruments necessarily lead to different computations and definitions of the aerosol layer
height, e.g. a weighted extinction height from a lidar extinction profile (e.g. Koffi et al., 2012), aerosol top height for stereoscopy

or aerosol layer effective height to characterise light penetration depth (e.g. Kim et al., 2024). In this paper, we focus on ALH from the TROPOspheric Monitoring Instrument (TROPOMI), which is computed assuming a geometrically thin layer of aerosols with a fixed thickness and variable altitude, and considered a centroidal height.

To validate the TROPOMI ALH, it is compared to CALIPSO/CALIOP L2 weighted extinction heights for clear-sky scenes. The first comparisons of TROPOMI ALH with CALIOP weighted extinction heights were provided by Nanda et al. (2019), showing large discrepancies in the retrievals over land. Tests of the same processor used on GOME-2 data (Nanda et al., 2018a) already indicated the large error sources that can be expected over bright surfaces. In the $O_2$ A-band, spectral points that represent photons which are less absorbed by oxygen, i.e. those which travel through the atmosphere most easily, have the lowest relative error. This favours spectral points that are affected more by surface reflection compared to spectral points that are affected more by aerosol scattering. Nanda et al. (2018b) proposed a dynamical (i.e. scene and wavelength dependent) reversal of this preference in the retrieval, but failed to yield significant improvements over land surfaces. In cases where the contribution of the surface to the top-of-atmosphere (TOA) reflectance signal is low and dominated by photons reflected by the aerosol layer, the TROPOMI aerosol layer height can be expected to be well retrieved. Indeed, Griffin et al. (2020) showed that TROPOMI ALH retrievals increasingly matched CALIOP average top/base heights for increasing geometric aerosol plume thicknesses and decreasing surface albedos. The mean differences in plume altitude ranged from an underestimation of TROPOMI ALH of more than 2 km for geometrically thin plumes to just 50 m for plumes thicker than 3 km. Similarly, the lowest difference was found for the lowest surface albedos.

The effect of the surface albedo is smaller in the $O_2$ B-band, especially for vegetated surfaces. Chen et al. (2021) showed that including the $O_2$ B-band improved the TROPOMI retrieval considerably over land. Improved surface reflectivity representation also decreases the error in the retrieved aerosol layer height, which is shown in this paper in section 3.1. However, for very bright surfaces the signal at TOA is still dominated by the surface and the retrieval remains challenging. In particular, for surface albedos at which the TOA reflectance does no longer depend on the aerosol optical thickness, the retrieval of aerosol optical thickness can have large biases (Seidel and Popp, 2012). To investigate if the same holds for the retrieval of aerosol layer height, Sanders et al. (2015) showed that derivatives of the reflectances with respect to aerosol layer height and surface albedo are different in the $O_2$ A-band, even for combinations of surface albedos and aerosol optical thicknesses that have the same reflectance in the continuum. This can be used with instruments that resolve the $O_2$ A-band spectrally to fit both the surface albedo and the aerosol layer height with an optimal estimation (OE) routine, since this relies on the derivatives of the parameters in the feature vector to compute the next step in each iteration. In this paper, it is demonstrated that this can be used to reduce the error in the ALH retrievals over land, by including the surface albedo in the OE feature vector, i.e. fitting the surface albedo along with the aerosol optical thickness and layer height. The presented surface albedo mitigation strategy will also be used for the Sentinel-5 successor series (Gühne et al., 2017), for which the $O_2$ B-band is not sampled at the same spatial resolution as TROPOMI, and for the geostationary Sentinel-4 UVN instrument (Stark et al., 2013), which lacks an $O_2$ B-band altogether.

The TROPOMI instrument and the operational ALH algorithm are introduced in section 2.1, along with the optimal estimation formalism (section 2.3). Then, the latest improvements in the ALH product are described. CALIOP weighted extinction

heights are used for verification. A set of cases with collocated CALIOP and TROPOMI measurements were selected, covering ocean and land and different aerosol types, described in section 2.4. Several improvements are treated. First, the introduction of the latest Directional Lambertian-Equivalent Reflectivity (DLER) surface albedo database, based on five years of TROPOMI data (section 3.1) is described. This improved the retrievals over land considerably compared to version 1 ALH data, which were biased strongly towards the surface. However, the improvement was not satisfactory, therefore the improvements that can be gained from fitting of the surface albedo are shown for land surfaces (section 3.2). Over ocean, the fitting of the surface albedo does not necessarily improve the retrievals and often even decreased its accuracy. A different fitting configuration was needed over water surfaces to get similar or improved results as before, which is described in section 3.3. The conclusion and recommendations from this study are presented in section 5.

## 2 TROPOMI Aerosol Layer Height algorithm description

The Aerosol Layer Height (ALH) from TROPOMI described in this paper refers to the operational, offline (as opposed to near-real time) data from the L2__AER_LH product, developed at the Royal Netherlands Meteorological Institute (KNMI) and distributed by the European Space Agency (ESA), as described in de Graaf et al. (2024). It is part of the aerosol product suite for TROPOMI, which further consists of the Absorbing Aerosol Index (AAI) at three wavelength pairs (335/367, 340/380, and 354/388 nm) from the L2__AER_AI product and the Aerosol Optical Thickness (AOT) at five wavelengths (354, 388, 416, 440, and 494) from the L2__AER_OT product.

It is noted that several alternative ALH retrievals are currently based on TROPOMI L1b data, employing different retrieval techniques or variations (e.g. Chen et al., 2021; Rao et al., 2022; Kim et al., 2024; Litvinov et al., 2024). The accuracies and sensitivities of those retrieval algorithms do not necessarily apply to the ALH product described here.

### 2.1 TROPOMI

TROPOMI (Veefkind et al., 2012) is a hyperspectral push-broom imaging spectrometer, launched on 13 October 2017 onboard the Sentinel-5 Precursor (S5P) satellite into a near-polar, Sun-synchronous orbit with a local equator crossing time of 13:30 for the ascending node, at an average altitude of 824 km above the Earth's surface. It observes reflected sunlight in the ultraviolet and visible wavelength range from 267–499 nm, the near-infrared (NIR) wavelength range from 661–786 nm, and the shortwave infrared wavelength range from 2300–2389 nm. For the ALH retrieval the NIR range is used, which has a spectral resolution of about 0.38 nm with a spectral sampling interval of 0.12 nm. The spatial resolution is around 5.5 km $\times$ 3.5 km at nadir, and the swath width about 2600 km across track, resulting in almost daily coverage of the global (sunlit) atmosphere.

### 2.2 Aerosol Layer Height product

The TROPOMI ALH retrieval algorithm matches TROPOMI reflectance measurements in the $O_2$ A-band with simulated reflectances, provided by a Neural Network (NN) that was trained on $1.6 \cdot 10^6$ randomly selected, simulated TROPOMI scenes. The first version of this NN was described in Nanda et al. (2019). The simulations were performed with the radiative trans-

125 fer model (RTM) "Determining Instrument Specifications and Analyzing Methods for Atmospheric Retrieval" (DISAMAR) (de Haan et al., 2022). In the $O_2$ A-band, absorption cross sections for absorbing molecules are modeled using a Voigt profile and line parameters from the HITRAN 2008 database (Rothman et al., 2009). Line mixing is taken into account for $O_2$.

The aerosol profile is modeled as a single aerosol layer containing weakly absorbing aerosols with a varying aerosol optical thickness and varying altitude. The angular distribution of the scattering of the light is described using a Henyey-Greenstein
function (Henyey and Greenstein, 1941), with an asymmetry parameter of 0.7. The single scattering albedo of the aerosols is 0.95, which is based on global long-term AERONET observations (de Leeuw et al., 2015). This model does not account for different aerosol types, but the ALH was shown to be robust with respect to fixed aerosol model parameters (Sanders et al., 2015; Nanda et al., 2019). The main reason is that differences between the modeled and the measured reflectances are mostly absorbed by the AOT, which is primarily controlled by the fit of the spectra in the continuum. Therefore, AOT is considered
an effective quantity and not to be used as an AOT measurement. On the other hand, the ALH is optimized in the retrieval and considered the prime retrieval target. Currently, no dynamic information (daily measurements) on aerosol type is available, but this may change with missions like Cloud, Aerosol and Radiation Explorer (EarthCARE), Plankton, Aerosol, Cloud, ocean Ecosystem (PACE) and MetOp-Second Generation Program A (MetOp-SG A), in which case a fit with different aerosol models may be considered for operational processing.

The algorithm follows the descriptions by Nanda et al. (2019, 2020), with a few notable exceptions. First, not only scenes with absorbing aerosols are processed, but all TROPOMI cloud-free scenes. The cloud screening is performed with the Suomi National Polar-orbiting Partnership (SNPP) Visible / Infrared Imaging Radiometer Suite (VIIRS) Enterprise Cloud Mask (ECM) (Kopp et al., 2014), reprojected onto the TROPOMI footprint. VIIRS flies in close formation with TROPOMI with a time difference of about 3–4 minutes. Second, instead of a constant pressure difference between the top and bottom of the
simulated aerosol layer, the geometric thickness of the layer is kept fixed at 250 m at each pressure level. The ALH algorithm still computes the top and bottom of the aerosol layer, and the average mid level, using pressure as the independent height variable at each iteration. However, pressure is converted to altitude assuming hydrostatic equilibrium, and the new layer top and bottom pressures are computed assuming the fixed geometrical thickness of the layer. This avoids semi-infinite layers at low pressure levels which occurred in previous versions, where a constant layer pressure difference of 50 hPa was assumed.
Lastly, the surface albedo is treated differently compared to previous versions, which is described in the remainder of this paper.

TROPOMI data are quality-controlled using a continuous quality assurance (QA)-value between 0, indicating non-converging retrievals or retrievals resulting in an error, and 1, indicating successful, non-compromised retrievals. QA-values below 1 indicate reduced quality due to possible issues, indicated by raised warning flags. Retrievals with QA-values 0.5 and below are
155 considered compromised and not recommended for use. Furthermore, users of the ALH are advised to be cautious with retrievals for scenes with low aerosol load and very bright surfaces. Results that were obtained for scenes with AOT lower than 0.3 (now indicated by a QA-value of at most 0.5) and surface albedo values above 0.4 were filtered in this paper, unless stated otherwise. The TROPOMI ALH data used in this paper refer to versions 02.04.00, released in July 2022, and version 02.08.00, released in November 2024, as indicated consistently throughout the paper.

## 2.3  Optimal Estimation

The retrieval of the TROPOMI ALH is based on the optimal estimation formalism described in Rodgers (2000), minimising a cost function $\chi^2$ that is given by

$$\chi^2 = [\mathbf{y} - \mathbf{F}(\mathbf{x}, \mathbf{b})]^T \mathbf{S}_\epsilon^{-1} [\mathbf{y} - \mathbf{F}(\mathbf{x}, \mathbf{b})] + (\mathbf{x} - \mathbf{x_a})^T \mathbf{S_a}^{-1} (\mathbf{x} - \mathbf{x_a}). \tag{1}$$

The first term represents a linear least-squares cost function, in which $\mathbf{y}$ are the measurements, a vector of measured reflectances for the different wavelengths in the $O_2$ A-band. The reflectance is the quotient of the upwelling radiance $I(\lambda)$ and the downwelling solar irradiance $E_0(\lambda)$, $R = \pi I(\lambda)/\mu_0 E_0(\lambda)$, where $\mu_0$ is the cosine of the solar zenith angle $\theta_0$. The forward model $\mathbf{F}(\mathbf{x}, \mathbf{b})$ consists of a vector of simulated reflectances, calculated by the ALH algorithm for a set of model parameters $\mathbf{b}$ and the state vector $\mathbf{x}$. $\mathbf{S}_\epsilon$ is the error covariance matrix of the measurements. The measurement errors are assumed to be independent, and therefore $\mathbf{S}_\epsilon$ is diagonal. Measurement error estimates can be determined from L1B radiance and irradiance noise estimates, but given the simplification of the forward model and limitations in the noise estimates, this can be expected to leave too little margin for convergence in many scenes. Therefore, a maximum value on the signal to noise ratio on the reflectance, called a 'noise floor', was introduced to create a margin for error contributions that are not taken into account in the normal error propagation. A good increase in convergences for TROPOMI ALH was found with a noise floor of 100.

The second term in equation 1 contains *a priori* information in order to constrain possible solutions, ensuring that the solution does not differ too strongly from the prior knowledge. $\mathbf{x_a}$ is the *a priori* state vector and $\mathbf{S_a}$ its associated error covariance matrix. The state vector elements are scaled with $\mathbf{S}_\epsilon$ to bring them in a range that increases the numerical stability, an operation called pre-whitening. For such a non-linear system with regularisation, the update of the state vector for each iteration $i$ can be found using the Gauss-Newton method by

$$\mathbf{x}_{i+1} = \mathbf{x_a} + (\mathbf{K}_i^T \mathbf{S}_\epsilon^{-1} \mathbf{K}_i + \mathbf{S_a}^{-1})^{-1} \mathbf{K}_i^T \mathbf{S}_\epsilon^{-1} [\mathbf{y} - \mathbf{F}(\mathbf{x_i}) + \mathbf{K}_i(\mathbf{x}_i - \mathbf{x_a})], \tag{2}$$

where the Jacobians $\mathbf{K}_i = \mathbf{K}(\mathbf{x}_i)$ are the derivatives $K_{ij} = \partial R(\mathbf{x}_i)/\partial x_{ij}$ for each of the state vector elements $x_{ij}$. The iterations can be started at $\mathbf{x}_0 = \mathbf{x_a}$. Since the linearisation point for the non-linear equation is $\mathbf{x} = \mathbf{x}_i$, which changes for each iteration, the simulated reflectances $\mathbf{F}(\mathbf{x}_i, \mathbf{b})$ and the Jacobians $\mathbf{K}_i$ have to be calculated at each iteration. Derivatives can be obtained from the reflectance NN directly, but because of the non-linearity, separate NNs were created to compute the derivatives with respect to each state vector element, to improve the accuracy of the operational ALH algorithm. In section 3.2 the extension of the forward model in which the surface albedo is included in the state vector is described. The retrieval is said to be converged to a solution when the state vector update is lower than the expected precision.

The weight of the prior information versus the weight of the measurements is determined by the covariance error matrices $\mathbf{S}_\epsilon$ and $\mathbf{S_a}$. The *a posteriori* maximum likelihood estimate of the state vector is given by (e.g. Rodgers, 2000)

$$(\mathbf{x} - \mathbf{x_a}) = \left[ \mathbf{K}^T \mathbf{S}_\epsilon^{-1} \mathbf{K} + \mathbf{S_a}^{-1} \right]^{-1} \mathbf{K}^T \mathbf{S}_\epsilon^{-1} (\mathbf{y} - \mathbf{K}\mathbf{x_a}). \tag{3}$$

**Table 1.** Description of selected cases of aerosol plume events, the difference in overpass times between Sentinel-5p and Earth-CARE and the surface under the track.

| Date | Location | | Description | Time diff. | Surface |
|------|----------|--|-------------|------------|---------|
| **CALIPSO** | | | | | |
| 1 Jun. '18 | $5°$–$25°$N,$10°$–$30°$W | Sahara, Northeast Atlantic | Dust | 11 min. | ocean |
| 31 Jul. '18 | $5°$–$25°$N,$40°$–$70°$E | Arabian Peninsula, Arab Sea | Dust | 32 min. | land & ocean$^\diamond$ |
| 15 Aug. '18 | $25°$S–$5°$N,$10°$W–$30°$E | Africa, Southeast Atlantic | Smoke | 1 min. | land & ocean |
| 21 Feb. '19 | $17.5°$–$30°$N,$80°$–$95°$E | India, Bay of Bengal | Anthropogenic pollution | 62 min. | land & ocean |
| 8 Jul. '19 | $55°$–$70°$N,$140°$–$170°$W | Alaska | Multiple smoke layers | 8 min. | land & ocean$^\diamond$ |
| 12 Feb. '20 | $30°$–$45°$N,$110°$–$125°$E | Asia | Anthropogenic pollution | 13 hours$^\mathbb{C}$ | land |
| 17 Jun. '20 | $0°$–$30°$N,$40°$W–$5°$E | Sahara, Northeast Atlantic | Dust | 55 min. | ocean$^\diamond$ |
| 7 Sep. '20 | $30°$–$46°$N,$105°$–$130°$W | North America | Multiple smoke layers | 25 min. | land and ocean$^\diamond$ |
| 19 Apr. '23 | $10°$–$45°$N,$95°$–$122.5°$W | China | Anthropogenic pollution | 134 min. | land |
| **EarthCARE** | | | | | |
| 26 Feb. '25 | $0°$–$20°$N,$5°$–$30°$W | Sahara, Northeast Atlantic | Dust | 4 min.$^\downarrow$ | land and ocean |

$^\mathbb{C}$ night-time overpass

$^\diamond$ used in section 3.3

$^\downarrow$ descending orbit

If no prior information is available, i.e. $\mathbf{S_a} \to \infty$, and $\mathbf{S}_\epsilon$ is diagonal with all diagonal elements having the same magnitude, this reduces to the linear system. Hence, the *a priori* error covariance matrix $\mathbf{S_a}$ can be used to allow more or less weight to the prior information. In section 3.3 the effect of different weights in the *a priori* error covariance matrix is described for retrievals over ocean.

## 2.4   Comparison with CALIPSO/CALIOP

For the validation of the ALH product, a set of nine different cases were selected where TROPOMI and CALIOP measurements were collocated. CALIOP is part of the CALIPSO satellite payload, which was part of the A-train constellation (Stephens et al., 2002), consisting of several satellite platforms flying in constellation in a polar orbiting, Sun-synchronous orbit, with an equator-crossing time similar to S5P. Therefore, the time difference between CALIOP and TROPOMI is small, generally less than one hour.

Table 1 lists the characteristics and details of the selected cases. They were selected to cover the time span of the TROPOMI mission, ocean and land surfaces, CALIOP measurements with various time differences, and events with different aerosol plumes, including desert dust, smoke from vegetation fires and industrial pollution. Note that the ALH retrieval does not take different aerosol types into account, but assumes weakly absorbing aerosols, as explained in section 2.2.

CALIOP L1 backscatter at 532 nm (V4-51) are used to illustrate the vertical backscatter cross section of the atmosphere along a CALIPSO track. L2 aerosol extinction profiles (V4-51), averaged over $0.15°$ latitude along the track, are used to compute the weighted extinction height, following (Nanda et al., 2020)

$$z_{\text{ext}} = \frac{\sum\limits_{k=1}^{n} \alpha_{\text{ext},k} \cdot z_k}{\sum\limits_{k=1}^{n} \alpha_{\text{ext},k}} \, , \tag{4}$$

where $z_k$ is the height from sea level in the $k^{\text{th}}$ lidar vertical level (in km), and $\alpha_{\text{ext},k}$ is the averaged aerosol extinction coefficient (in km$^{-1}$) at the same level.

## 3 Aerosol Layer Height product improvements over land

### 3.1 TROPOMI DLER surface albedo database

In July 2022, version 02.04.00 of TROPOMI L2 data was introduced, based on an improved calibration of L1b data. All L2 data up to that date were reprocessed replacing version 1 data. Additionally, a new DLER climatology based on TROPOMI measurements (Tilstra et al., 2024) was introduced, to replace a DLER climatology based on GOME-2 measurements that was used in the initial period of the TROPOMI mission when enough TROPOMI measurements were not yet available. The database contains the TROPOMI surface DLER retrieved for 21 wavelength bands outside atmospheric absorption bands, ranging from 328 to 2314 nm with a spatial resolution of $0.125° \times 0.125°$, which is an improvement over the GOME-2 DLER climatology. However, the most important aspect is that the DLER from TROPOMI measurements is based on the correct afternoon solar-viewing geometry for TROPOMI, whereas the GOME-2 climatology is based on a morning geometry. Therefore, the directionality of the DLER from the GOME-2 climatology was unsuitable and only the non-directional LER value could be used. Since version 02.04.00 the actual directional DLER values from the climatology are used. The DLER is updated regularly, the first version of this DLER climatology was based on three years of TROPOMI data, the current version is based on five years of TROPOMI data.

The use of the DLER surface albedo improved the retrievals compared to previous versions. Figure 1 shows a histogram of differences for 10 months of collocated CALIOP weighted extinction heights and TROPOMI ALHs, over land and ocean surfaces. The CALIOP weighted extinction heights were the same as used by Nanda et al. (2020), who compared these data with collocated TROPOMI version 01.03.00 data for the same period. Figure 1 can be compared to their Figure 2, except for two differences: In version 1, ALH was only retrieved for AAI>1, to ensure (absorbing) aerosol plumes. Since version 2, all cloud-free pixels are processed. Also, slightly different collocation methods were used. Nanda et al. (2020) applied a nearest neighbour approach to match all CALIOP profiles to the nearest TROPOMI pixel, resulting in $1.5 \cdot 10^6$ collocations. Here, a CALIOP profile is averaged over $0.15°$ latitude along the CALIPSO track and matched with the average TROPOMI ALH in

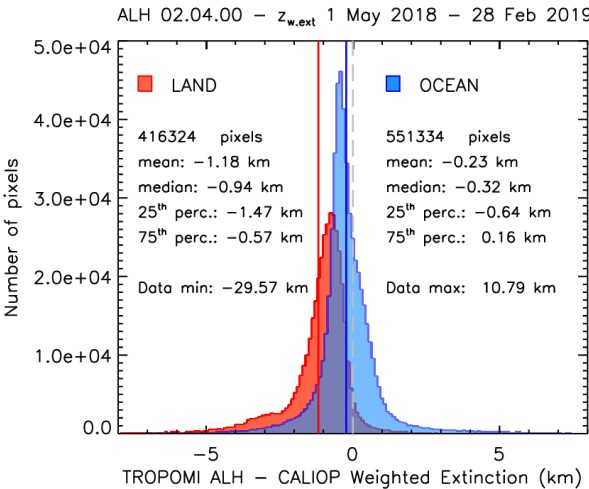

**Figure 1.** Histogram of differences between TROPOMI ALH version 02.04.00 and CALIOP weighted extinction height from co-located data between 1 May 2018 and 28 February 2019. The blue histogram represents TROPOMI pixels over the ocean, whereas the red histogram is for TROPOMI pixels over land. The blue line represents the mean difference between TROPOMI ALH and CALIOP weighted extinction height over the ocean, and the red line represents the mean difference over land. The pixels were filtered for QA-values greater than 0.5, AOT greater than 0.3 and surface albedo lower than 0.4. This discarded $1.7 \cdot 10^6$ points or 63% of the pixels.

a 40 km radius, resulting in a total of $1.2 \cdot 10^6$ collocations. The statistics are quite different for the two versions of TROPOMI ALH. Nanda et al. (2020) found a mean difference between TROPOMI ALH and CALIOP weighted extinction height of

$-2.4$ km over land and $-1.0$ km over ocean. For version 02.04.00 the differences are $-1.2$ km over land, and $-0.2$ km over ocean, respectively. Clearly, a more appropriate DLER database is important for an accurate ALH retrieval.

     An example of the TROPOMI ALH version 02.04.00 over North America is shown in Figure 2a, for a complicated scene with multiple layers of smoke from California and Oregon on 7 Sept. 2020 over the varying terrain of northern America. This severe smoke event was caused by a series of mega-fires along the United States west coast, ignited by thunderstorms following

seasonal dry periods. The retrieval shows ALH values close to the surface and many open parts where the algorithm did not converge. The thickest part of the plume (around $37°$N,$116°$ W) was not retrieved. In Figure 2b CALIOP measurements along the track in the upper plot is shown, an attenuated backscatter curtain plot with the weighted extinction height from CALIOP extinction profiles overplotted as purple dots. The collocated, averaged TROPOMI ALH along the track is overplotted as orange squares. According to the CALIOP extinction profiles, the plume height is generally at a higher altitude than indicated

by the TROPOMI ALH, which is biased strongly towards the surface. The highest CALIOP extinction heights are up 9 km in altitude, while the TROPOMI ALH never reaches higher than 3 km, if it is retrieved successfully at all.

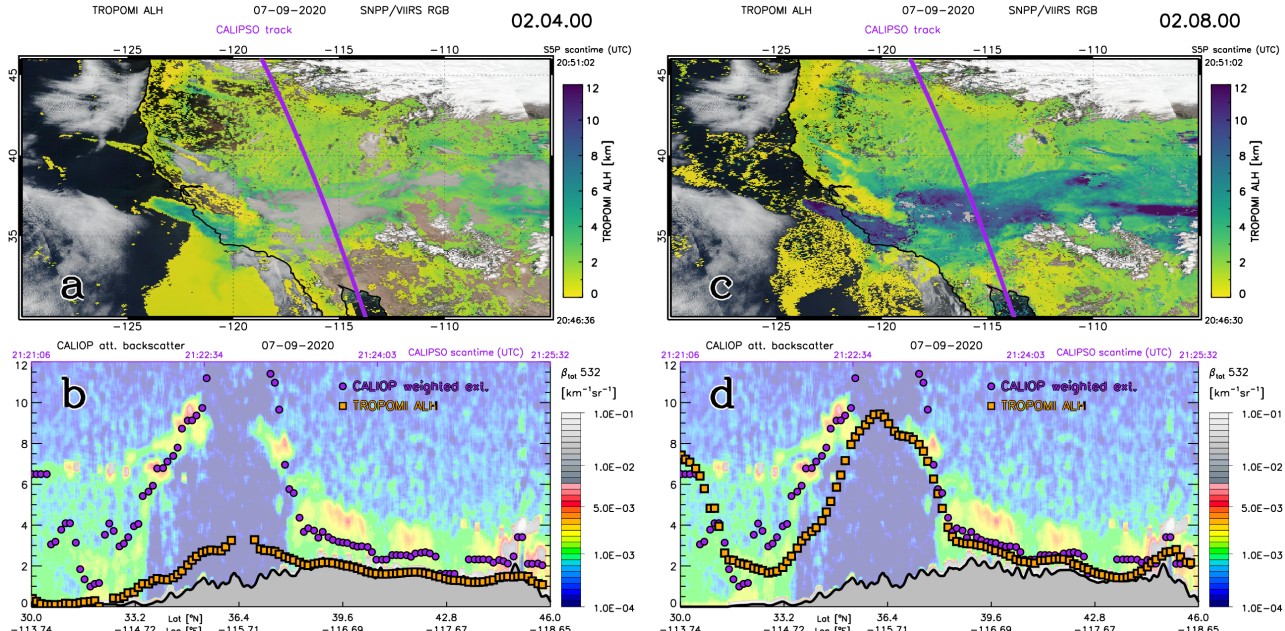

**Figure 2.** (a) True color (RGB) image from SNPP/VIIRS on 7 Sept. 2020 showing Californian smoke plumes over north-west America, overlaid with TROPOMI ALH, version 02.04.00 from 20:46:36–20:51:02 UTC. The purple line shows the daytime CALIPSO track over the area on the same day from 21:21:06–21:25:32 UTC. (b) CALIOP L1 532 nm attenuated backscatter curtain image along the purple track in the top panels, overlaid with the CALIOP weighted extinction height (purple dots) from L2 extinction profiles at 532 nm (averaged every 0.15° latitude along the track) and the average TROPOMI ALH of collocated pixels within a 0.5° radius of the CALIOP extinction profiles along the track. Note that ALH is given from sea level, so retrievals at the ground over an elevated surface are not necessarily zero. Elevated surfaces are grey shaded in the bottom curtain plots. (c) and (d): Same as (a) and (b) respectively, but with TROPOMI ALH version 02.08.00.

## 3.2 Surface albedo in feature vector

In order to improve the accuracy and convergence of the retrievals over land further, the fitting of the surface albedo in the inversion algorithm was introduced. The search for a minimization of the cost function in the inversion is guided by the derivatives with respect to the state vector elements. Since for the surface albedo two values are included in the state vector, one just below the $O_2$ A-band and the other just beyond it, two new NN had to be created. Therefore, the inversion algorithm now relies on five NNs, one for the reflectance, and four for the derivatives. The *a priori* correlation coefficient between the two surface albedo values was set to 0.9999, strongly connecting their values in the fit. Table 2 lists the state vector elements and the *a priori* values and errors for the versions used in this paper. The aerosol mid pressure is started 100 hPa above the surface and the aerosol optical thickness is started at 0.5. In version 02.04.00 the surface albedo is not fitted but DLER values at 758 nm and 772 nm are used. In version 02.08.00 these DLER values are used as *a priori* values. The closer the initial setting is to the actual value, the faster and more accurate the retrieval is.

**Table 2.** State vector elements and typical *a priori* values and errors for the Aerosol Layer Height retrieval algorithm. The version column states the lowest version, used in this paper, for which the state vector element is implemented.

| State vector element | | Symbol | v 02.04.00 | | v 02.08.00 | |
|---|---|---|---|---|---|---|
| | | | $\mathbf{x_a}$ | error | $\mathbf{x_a}$ | error |
| Aerosol mid pressure | | $p_{\mathrm{mid}}$ | $p_{\mathrm{surf}} - 100$ hPa | 200 hPa | $p_{\mathrm{surf}} - 100$ hPa | 200 hPa |
| Aerosol optical thickness at 760 nm | | $\tau_{\mathbf{0}}$ | 0.5 | 1.0 | 0.5 | 1.0 |
| Surface albedo at 758,772 nm | land | $\mathbf{A_s}$ | not fitted, DLER | | DLER | 0.05 |
| | ocean | $\mathbf{A_s}$ | | | DLER | 0.01 |

The effect of the surface albedo fitting is clearly demonstrated in Fig. 2c and d. The coverage of the ALH is much better, with the number of successful retrievals increasing from 50,237 to 74,847 for the area shown in Fig. 2. The high altitude plumes, which are also the thickest plumes, are much better captured. Fig. 2d shows that the high altitude plumes are now retrieved up to 10 km altitude, which is close to the CALIOP weighted extinction heights. Also, the retrievals over the mountains around 40°N close to the elevated surface are close to the surface but not at the surface, in line with what is expected from CALIOP weighted extinction heights.

### 3.3 Surface albedo fit over ocean

The inclusion of the surface albedo in the feature vector produced good results over land, but over ocean the results were not necessarily improved. In general, the surface albedo has a small influence on the aerosol layer height retrieval over the dark ocean surface. Only in sunglint regions the ocean surface albedo can generally become very high. These regions are flagged even though they may give reasonable results. Including the surface albedo in the fit sometimes resulted in wildly varying fitted albedos, when the OE procedure used the surface albedo to compensate for uncertainties in the aerosol optical thickness or aerosol layer height. Therefore, over the oceans the fitting range of the surface albedo was limited by limiting the *a priori* error in the OE for ocean retrievals. For land surfaces, the *a priori* error was set to 0.05, allowing for a wide range of land surface albedos. Over ocean, the optimal error setting was determined using a test on four of the cases with collocated CALIOP measurements over ocean.

The *a priori* errors for retrievals over ocean were varied from a very small number (0.002) to the same number as for land surfaces (0.05). In the first case, the setting is so tight that the OE method can be considered not fitting the surface at all, but using the surface albedo from the DLER database as it is. In the latter case, the setting is so relaxed that nonphysical surface albedos may be found to compensate for errors in the other two parameters. The optimal setting was determined by comparing the results with CALIOP weighted extinction heights for different settings and finding the optimal value at the point where the differences are small and close to the results over land. The test is illustrated in Fig. 3, which shows the mean and median difference between TROPOMI ALH and CALIOP extinction heights using different *a priori* error settings over ocean. For an *a priori* error of 0.002, the retrievals over ocean were on average $-0.94$ km from the CALIOP weighted extinction heights,

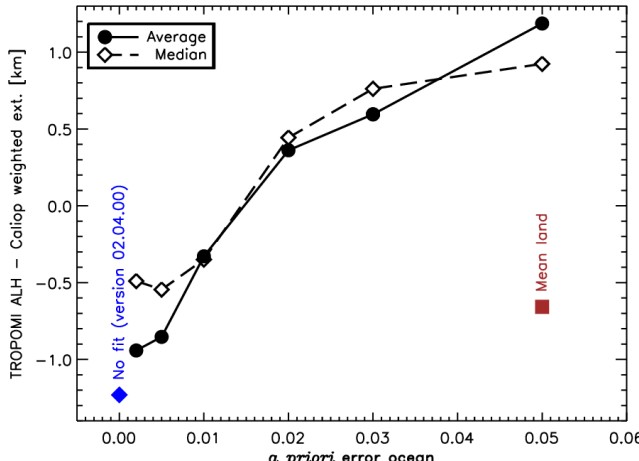

**Figure 3.** Sensitivity of the Aerosol Layer Height retrieval for the *a priori* error settings for retrievals over ocean. The solid line shows the average difference between TROPOMI ALH and CALIOP weighted extinction height for different values of the *a priori* error and the dashed line shows the median of the differences, determined using cases indicated in Table 1 by a '◇'. The *a priori* error for retrievals over land was always set to $0.05$, for which the average difference is about $-0.7$ km, indicated by the 'Mean land' square. In version 02.04.00 the surface albedo was not included in the fit and the difference between TROPOMI ALH and CALIOP weighted extinction height was about $-1.2$ km.

which is close to the unfitted results that are indicated in the figure (version 02.04.00 data). For a value of $0.05$, the retrievals over ocean are on average about $1.2$ km higher than the CALIOP weighted extinction heights. Over land, this same setting produces a much smaller difference, TROPOMI ALH being about $-0.7$ km from CALIOP (closer to the surface, indicated

by the 'Mean land' square). For a value of $0.01$ the average difference became $-0.33$ km, close to the difference over land surfaces, and this value for the *a priori* error over ocean was adopted for version 02.08.00 ALH (see Table 2). Note that, judging from the steep slope, the results are sensitive to the settings of the *a priori* setting in this range and the test data set is small.

  The effect of the settings change is illustrated in Fig. 4 in the same way as Fig. 2 but now for a case of desert dust over the Arabian Peninsula and the Gulf of Aden. This case is interesting because of the land-sea contrasts and the very bright

land surface. In Fig. 4a the ALH version 02.04.00 is shown, which clearly shows the problems with this version. At the top of the panel, over land, the desert dust plume is visible as a bright haze, which is not captured at all because none of the pixels converged in this area, mainly due to the bright surface. Over ocean, the convergence is quite good with a smooth ALH field, except for the sun glint region which is filtered. However, it is clear that the ALH retrieved over ocean is not continued over land. In Fig. 4b the comparison with the CALIOP weighted extinction is shown along the CALIPSO track in the top

panel, which shows the general large bias towards the surface for the average TROPOMI ALH, especially over land surfaces. In Fig. 4c and d, the same figures are shown but using ALH version 02.08.00. Over land, the desert dust plume is covered much better, with more retrievals converging successfully even over the brightest parts. Over ocean, the algorithm handles sun

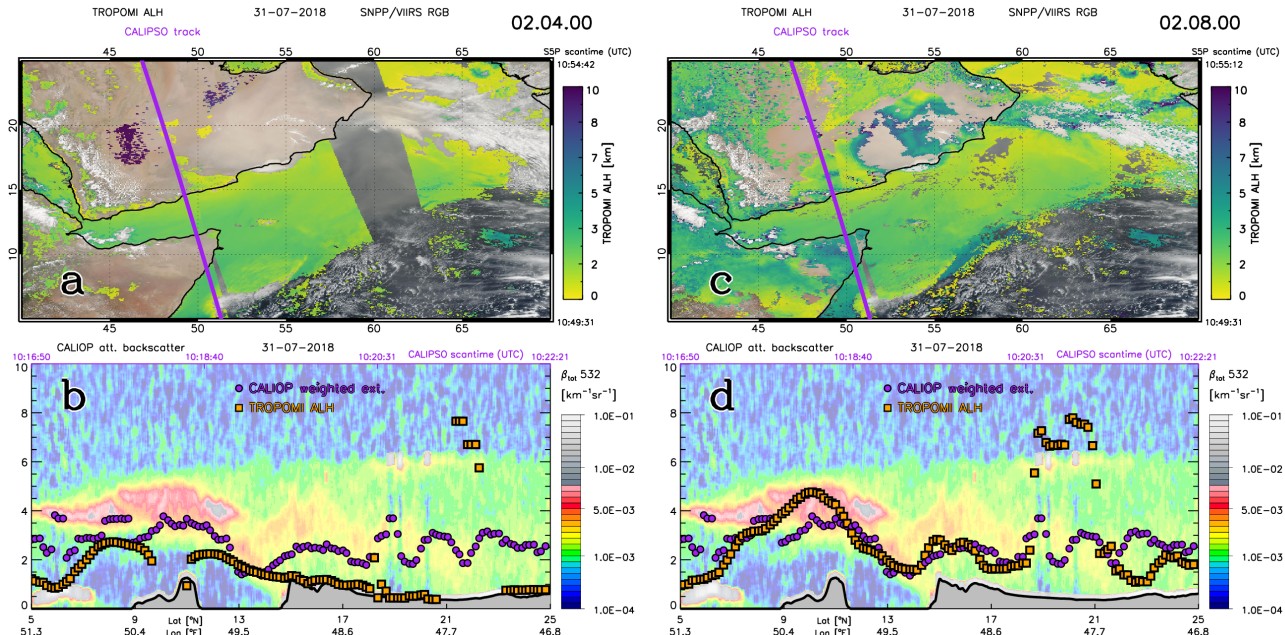

**Figure 4.** (a) True color (RGB) image from SNPP/VIIRS on 31 Jul. 2018 showing a dust plume over the Arabian Peninsula, overlaid with TROPOMI ALH, version 02.04.00 from 10:49:31–10:55:12 UTC. The purple line shows the daytime CALIPSO track over the area on the same day from 10:16:50–10:22:21 UTC. (b) CALIOP L1 532 nm attenuated backscatter curtain image along the purple track in the top panels, overlaid with the CALIOP weighted extinction height (purple dots) from L2 extinction profiles at 532 nm (averaged every $0.15°$ along the track) and the average TROPOMI ALH of collocated pixels within a $0.5°$ radius of the CALIOP extinction profiles along the track. (c) and (d): Same as (a) and (b), but with TROPOMI ALH version 02.08.00.

glint well, and these retrievals are retained in the product, albeit with a reduced QA-value of $0.7$ and a raised sunglint flag, to alert the user for possible problems related to sun glint. Also, the land-sea contrasts are small. Along the CALIPSO track, the

transition from land to sea is smooth and following the weighted extinction height by CALIOP closely. In general, the retrieval differences between the two instruments are small, except for a few regions that are probably cloud contaminated. E.g. around $20°$N the high ALH retrievals coincide with high backscatter values in the curtain plot around $6$ km, and around $5°$N the low ALH values coincide with high backscatter values around $0.5$ km. CALIOP extinction profiles are unaffected by these values if correct backscatter-to-extinction ratios and feature masks were applied.

Similar comparisons between CALIOP and TROPOMI ALH for the rest of the selected cases in Table 1 are presented in the appendix. The impact of the surface albedo fitting for all cases is given in Fig. 5. It shows the comparison of CALIOP weighted extinction height and TROPOMI ALH version 02.04.00 (left panels) and 02.08.00 (right panels). For version 02.04.00, the TROPOMI ALH is largely underestimated compared to CALIOP retrievals, and the correlation is poor over both ocean and land. The mean difference is about $-1.9$ km over land and $-0.9$ km ocean. This is different from Fig. 1 because of the

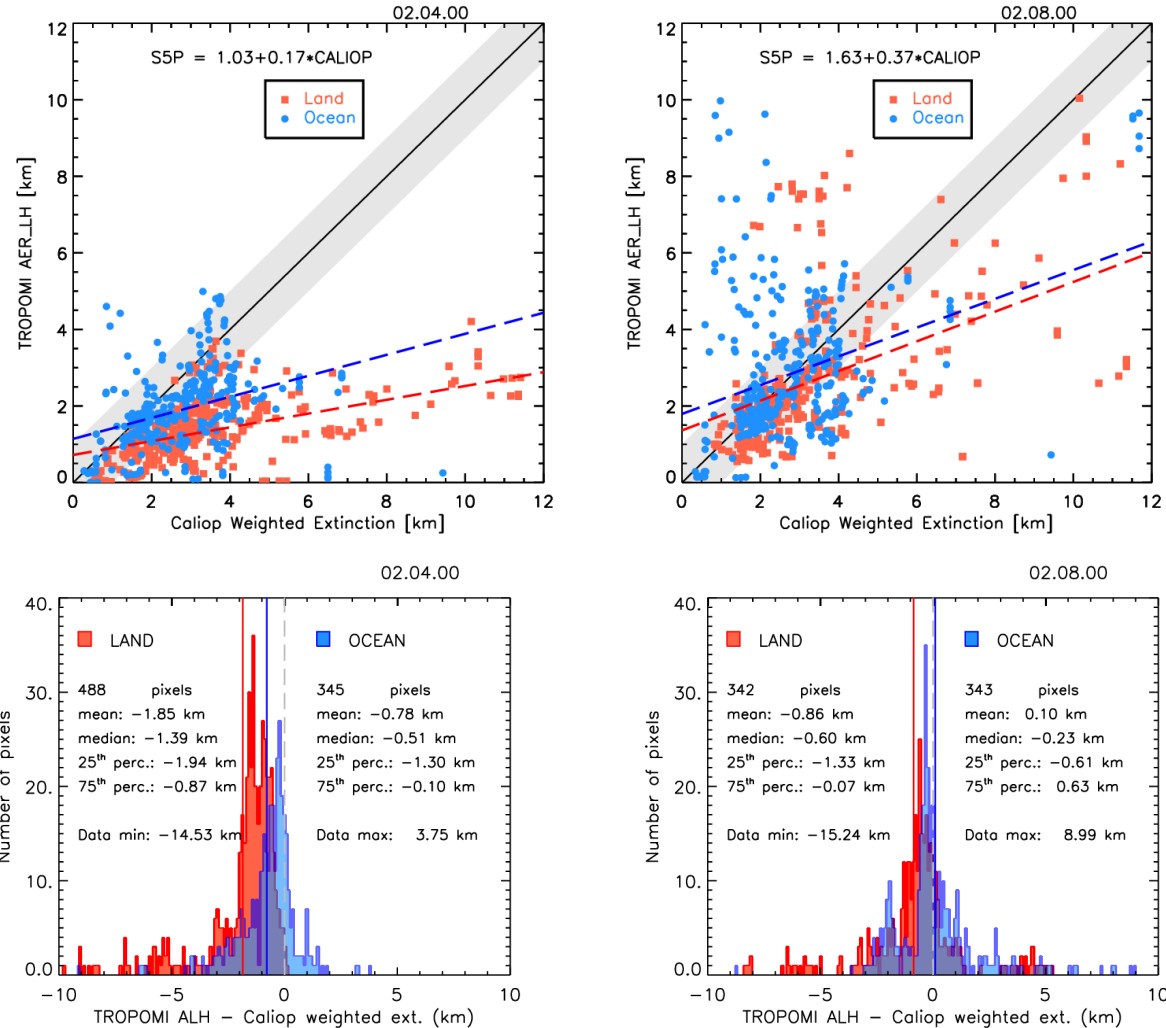

**Figure 5.** Upper panels: Comparison of CALIOP weighted extinction and TROPOMI ALH version 02.04.00 (left) and 02.08.00 (right) for the selected cases in Table 1 over land (red squares) and ocean (blue dots). A CALIOP profile was averaged each 0.15° latitude and compared with the TROPOMI ALH within a 0.5° radius. The black line shows the one-to-one line and the grey area the requirement for ALH of ±1 km. Lower panels: Histogram of differences between the points in the top panels. The blue line represents the mean difference between TROPOMI ALH and CALIOP weighted extinction height over the ocean, and the red line represents the mean difference over land. The pixels were filtered for AOT greater than 0.3 and surface albedo lower than 0.4.

low number of comparisons (743 pixel) and the selection of clear events with thick aerosol plumes. In the right panel, the comparison has improved considerably for version 02.08.00. The average difference is about −0.9 km over land and negligible average differences over ocean, albeit with a large spread. A clear correlation can be established between CALIOP weighted

extinction height and TROPOMI ALH version 02.08.00, with 57% of the points over land within the requirements of $\pm 1$ km and 59% of the pixels over ocean. These numbers were 30% over land and 65% over ocean for 02.04.00, respectively.

## 4  Comparison of version 02.08.00 with ATLID

The EarthCARE mission, launched on 28 May 2024, is a multi-instrument mission, operated jointly by ESA and JAXA. The ATmospheric LIDar (ATLID) is a three-channel, linearly polarized, high-spectral-resolution lidar (HSRL) system operating at 355 nm. L2a lidar extinction profiles from the extinction, backscatter, and depolarization (A-EBD) product (Donovan et al., 2024) were used to compute weighted extinction heights using Eq. 4, except at 355 nm.

Version 02.08.00 of the TROPOMI ALH was released in November 2024 and is currently available in near-real time, and in offline mode after two weeks of sensing. A desert dust outbreak on 26 February 2025 over the north-east Atlantic Ocean is used here to compare the latest version of TROPOMI ALH with measurements from ATLID.

Figure 6a shows two adjacent TROPOMI orbits with a small gap inbetween, visible in the middle of the plot, where TROPOMI orbits do not overlap near the equator. At the top of the panel, where the orbits do overlap, a very small east-west jump in the ALH can be observed. More interestingly, the transition from land to ocean in the ALH is almost negligible. The EarthCARE daytime track crosses the land-sea boundary several times, and no noticeable jump in the ALH is observed along this track (Fig. 6c). The comparison between ALH and the ATLID weighted extinction height is very good. Figure 6b shows the ATLID extinction profile for a location at the edge of the cloud-free part of the scene, illustrating the complicated profiles that result in an average weighted extinction height, and ALH, inbeteen the dust layer and a more elevated layer which is a cloud. In this case, ALH and weighted extinction height are close, but depending on the cloud filtering for ALH and the feature mask applied to ATLID data, the measurements can deviate more strongly, especially when the time difference between the measurements becomes larger.

## 5  Conclusions

The high spectral sampling of the $O_2$ A-band by spectrometers like TROPOMI allows the detection of the height of scattering layers even for weak scatterers like aerosols. This is generally challenging over bright surfaces, but the $O_2$ A-band contains information on the derivatives of the reflectance with respect to the aerosol layer height and surface albedo, that is used in an optimal estimation routine. The inclusion of the surface albedo in the OE fit showed a significant improvement in the TROPOMI ALH accuracies as quantified by collocated CALIOP weighted extinction heights. The use of a TROPOMI-based DLER surface albedo climatology improved the retrievals over land, on average from about $-2.4$ km lower than CALIOP weighted extinction heights to $-1.2$ km. By using the DLER values as *a priori* values for surface albedo fits in the OE routine, the inversions converged faster and the differences with CALIOP weighted extinction heights decreased to $-0.9$ km on average over land for a selected set of data. In addition, the coverage increased considerably, with about 1.5 times more successful convergences.

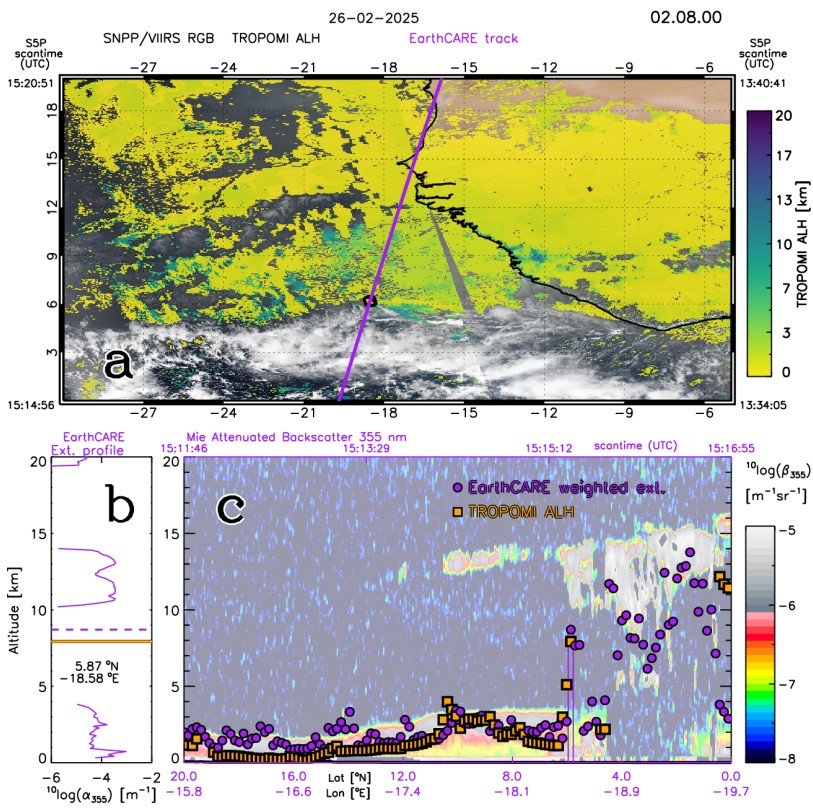

**Figure 6.** (a) True color (RGB) image from SNPP/VIIRS on 26 Feb. 2025 showing a desert dust outbreak over the Sahel and north-east Atlantic Ocean, overlaid with TROPOMI ALH, version 02.08.00 from 13:34:05–13:40:41 UTC (covering the right part of the image) and from 15:14:56–15:20:51 UTC (covering the left part of the image). The purple line shows the daytime EarthCARE track over the area on the same day from 15:11:46–15:16:55 UTC. Note that during daytime, EarthCARE is in a descending orbit, while Sentinel-5P is in an ascending orbit. The orange-black dot shows the location of the profile in panel (b); (b) ATLID L2 extinction profile at 335 nm at 5.9°N, -18.6°E (solid purple line), weighted extinction height of this ATLID profile (dashed purple line) and the TROPOMI ALH at this location (orange-black line); (c) ATLID L1 335 nm mie attenuated backscatter curtain image along the purple track in the top panel, overlaid with the ATLID weighted extinction height (purple dots) from L2 extinction profiles at 335 nm (averaged every 0.15° latitude along the track) and the average TROPOMI ALH of collocated pixels within a 0.5° radius of the ATLID extinction profiles along the track. The location of the profile in (b) is indicated by vertical purple lines.

A recent study on the Ocean and Land Colour Instrument (OLCI) onboard Sentinel-3 by the authors of the present paper confirmed the necessity of the high spectral resolution within the $O_2$ A-band to include the surface albedo in the OE fit. OLCI has three relatively broad (10-20 nm wide) bands within and two outside the $O_2$ A-band that can be used to retrieve ALH. This was first demonstrated in a dedicated study using a LookUp Table (LUT) approach by Jänicke et al. (2023). Application of the TROPOMI ALH retrieval algorithm confirmed the suitability of the OLCI $O_2$ A-band measurements for the same cases as presented in this study, but only over ocean and without fitting of the surface albedo. The retrievals over land were biased strongly towards the surface, by about 1–2 km. The reason is the low sensitivity of the derivatives with respect to surface albedo due to the low spectral information in the OLCI $O_2$ A-band measurements. Two approaches to improve this situation are currently under consideration: the application of a high spatial resolution Land Surface Reflectivity (LSR) climatology based on OLCI measurements, and the restoration of the high spectral information in the OLCI $O_2$ A-band, that is measured by the instrument but down-sampled in the processing.

The surface albedo fitting settings for TROPOMI were necessarily different over land and ocean surfaces. Over ocean, the surface albedo was prone to errors as compensation for inaccuracies in the aerosol optical thickness or aerosol layer height parameters. Limiting the *a priori* error to such a small value that the OE procedure effectively adopted the surface albedo value without fitting did not solve the problem. An optimal setting was found using a limited dataset and optimising the accuracies of the retrievals over ocean by varying the the *a priori* error. This procedure improved the surface albedo fitting over ocean, with an average difference with CALIOP weighted extinction heights of only about 0.1 km, and decreased the land-sea contrasts in the resulting ALH. It is noted here that the test is based on a limited dataset and is sensitive to differences in the *a priori* error setting. Ideally, results are not dependent on a prior setting, although the purpose of *a priori* information is exactly that: to stabilise the inversion and optimise the results.

The latest ALH retrievals were compared to ATLID weighted extinction heights that recently became available, in the same way as the test cases were compared with CALIOP weighted extinction heights. This independent verification showed also a very good comparison between the datasets, even though ATLID has different characteristics and measures at 355 nm. The land-sea contrast seems invariably low in the latest ALH data since the introduction of version 02.08.00 in November 2024. The suitability of the settings should be analysed again when a larger dataset is available. Then, more optimal settings may be applied in a reprocessing of the TROPOMI ALH dataset, but this is not foreseen until 2026 at the earliest.

The ALH algorithm is also being developed for Sentinel-4, to be launched in July 2025, and Sentinel-5, scheduled for launch in August 2025. For these instruments, such settings as the *a priori* error have to be analysed for each instrument individually. The presented criterion of a small difference between land and ocean is a good starting point, and can be done without a validation dataset.

## Appendix A: Comparison of cases

In this section, the TROPOMI ALH, both versions 02.04.00 and 02.08.00, are compared with CALIOP data for the cases listed in Table 1 that were not yet treated in the main text.

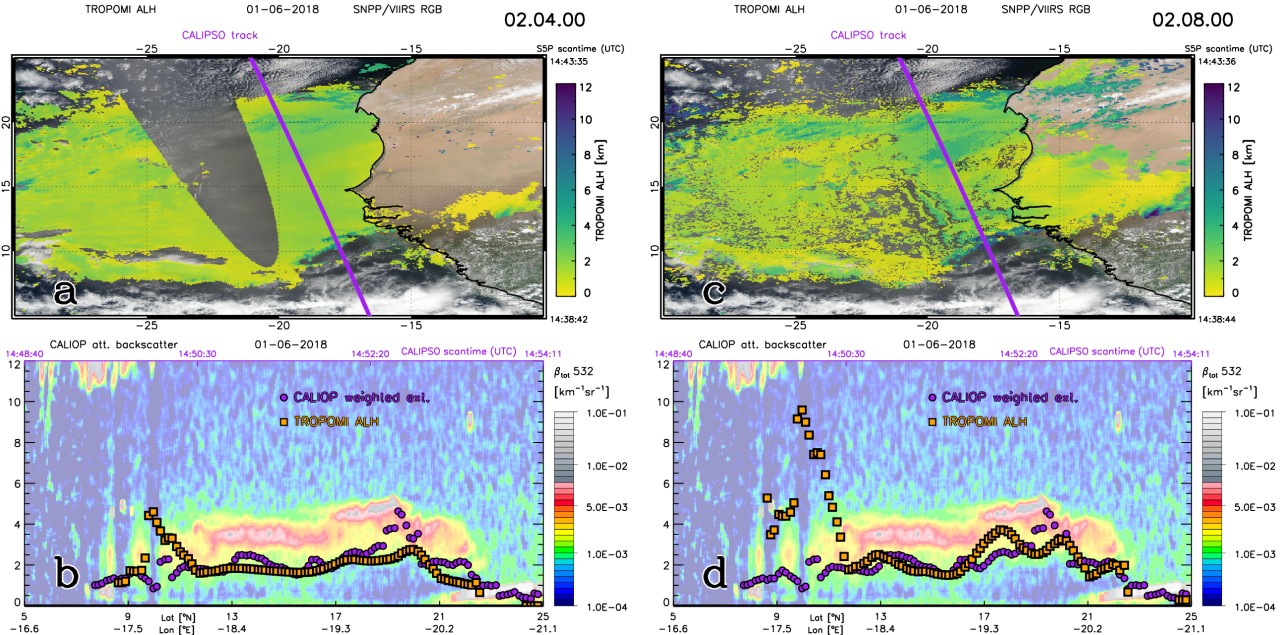

**Figure A1.** (a) True color (RGB) image from SNPP/VIIRS on 1 June 2018 showing a dust plume over the Sahara desert and the northeast Atlantic Ocean, overlaid with TROPOMI ALH, version 02.04.00 from 14:37:59–14:43:35 UTC. The purple line shows the daytime CALIPSO track over the area on the same day from 14:48:40–14:54:11 UTC. (b) CALIOP L1 532 nm attenuated backscatter curtain image along the purple track in the top panels, overlaid with the CALIOP weighted extinction height (purple dots) from L2 extinction profiles at 532 nm (averaged every $0.15°$ latitude along the track) and the average TROPOMI ALH of collocated pixels within a $0.5°$ radius of the CALIOP extinction profiles along the track. (c) and (d): Same as (a) and (b), but with TROPOMI ALH version 02.08.00.

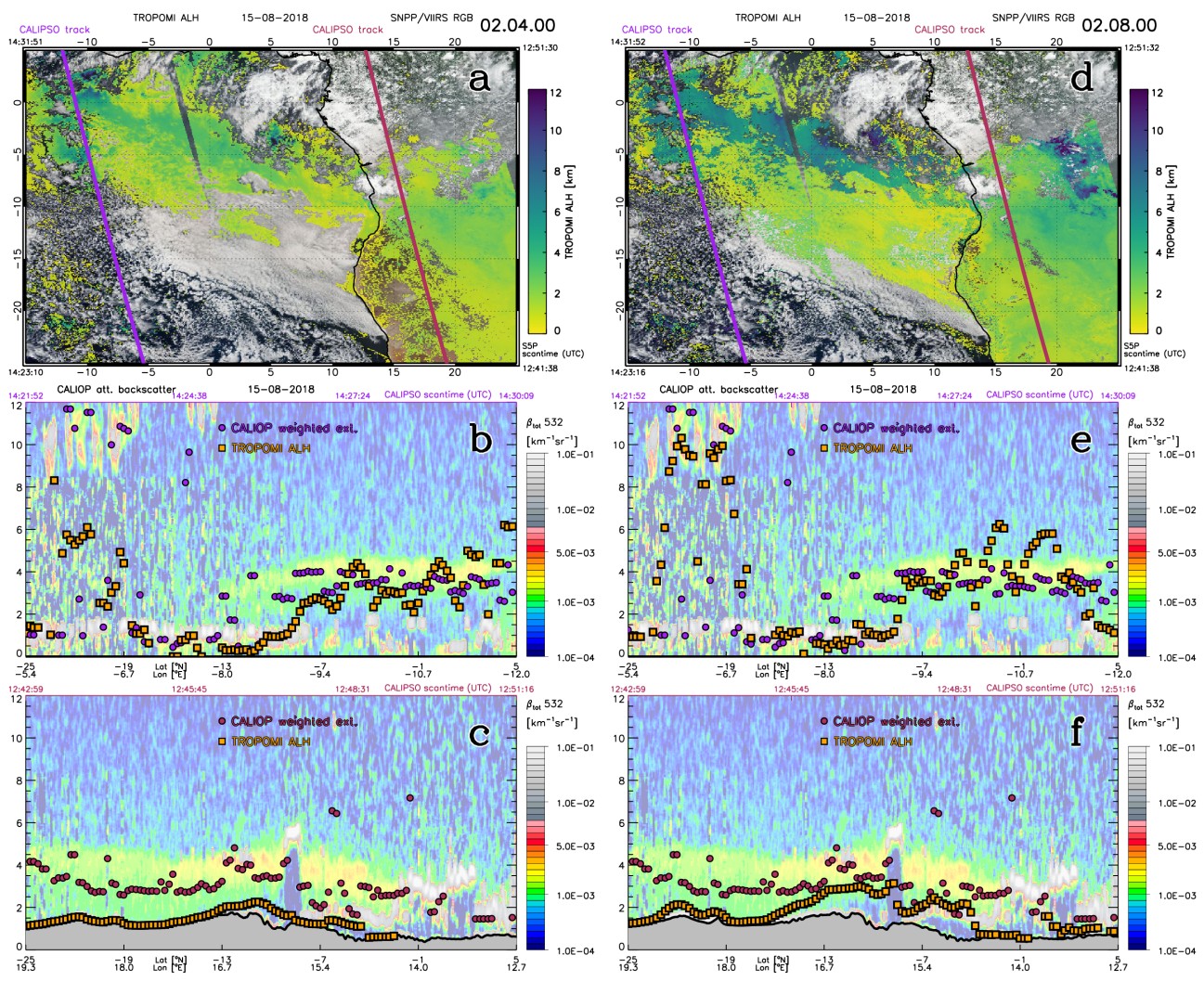

**Figure A2.** Same as Fig A1, but on 15 Aug. 2018 showing biomass burning smoke during two overpasses, the southeast Atlantic ocean from 14:23:52–14:31:51 UTC for TROPOMI data and 14:21:52–14:30:09 UTC for CALIOP data, and over central African and from 12:41:52–12:50:17 UTC for TROPOMI data and 12:42:59–12:51:16 UTC for CALIOP data.

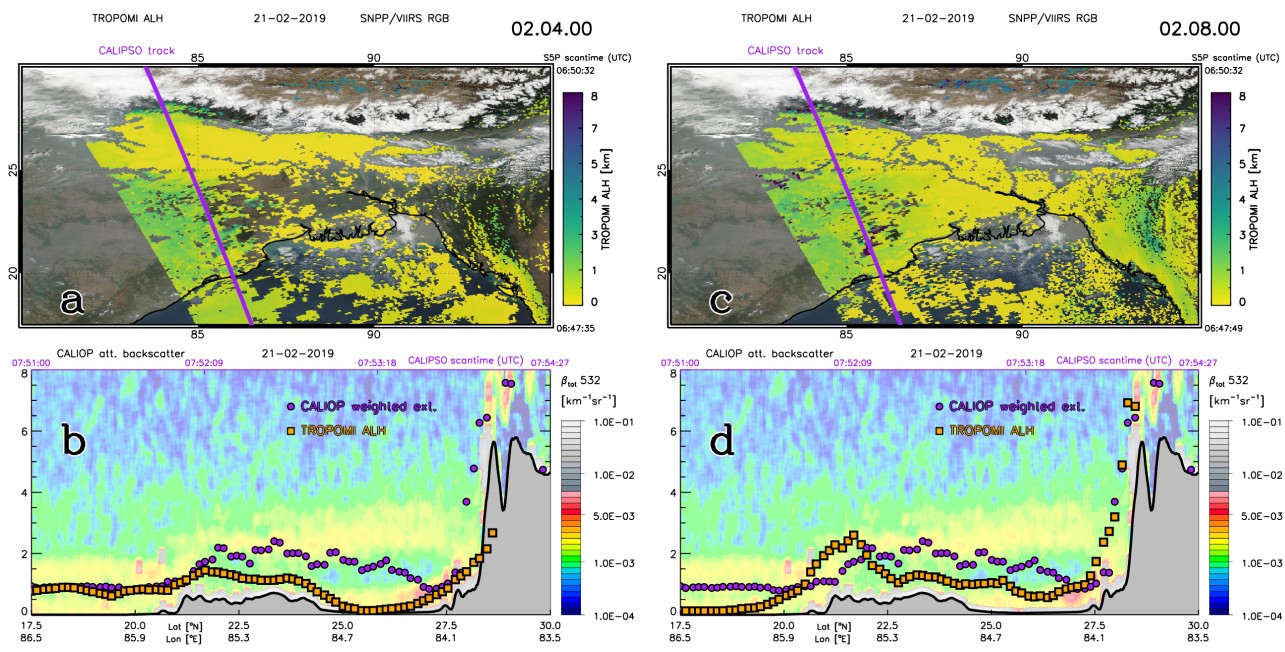

**Figure A3.** Same as Fig A1, but on 21 Feb. 2019, showing biomass burning smoke and industrial pollution south of the Himalayas, from 6:47:28–6:51:00 UTC for TROPOMI data and 7:50:52–7:54:27 UTC for CALIOP data.

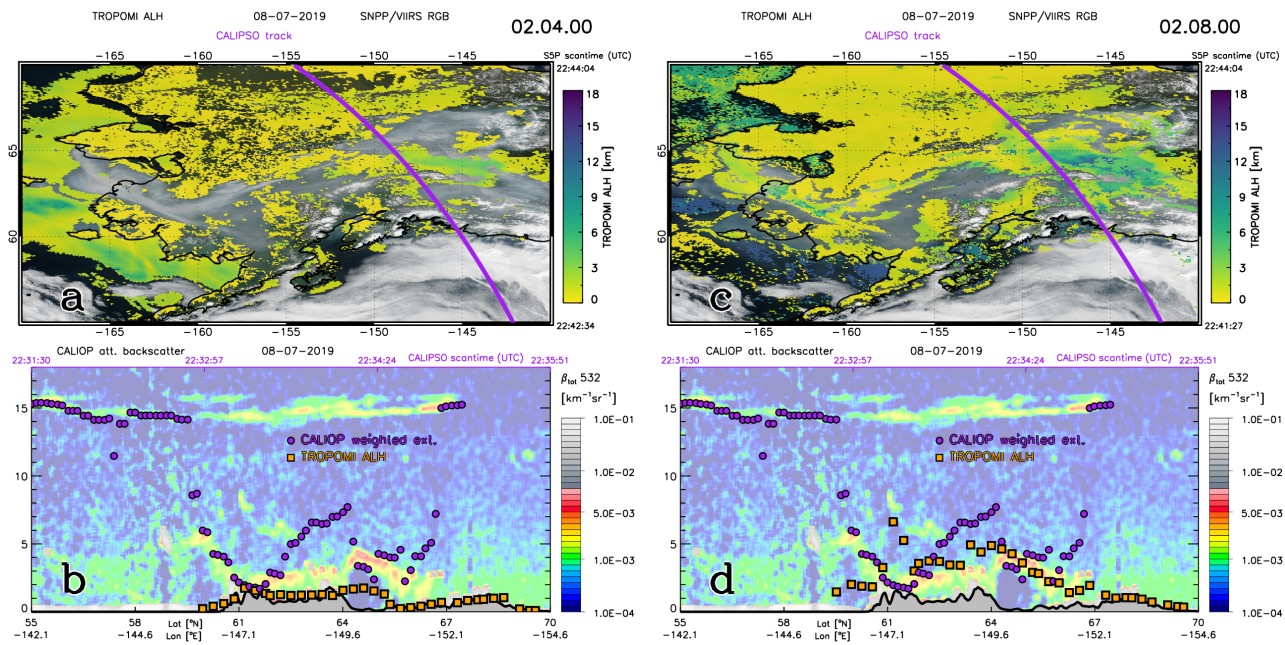

**Figure A4.** Same as Fig A1, but on 8 July 2019, showing multiple layers of biomass burning smoke over Alaska, from 22:41:13–22:43:07 UTC for TROPOMI data and 22:31:30–22:35:51 UTC for CALIOP data. CALIOP weighted extinction heights are found at around 15 km altitude around 56°N which coincides with a peak in the L1 backscatter coefficient due to a high altitude layer of smoke. The TROPOMI ALH is cloud-screened at those latitudes. TROPOMI ALH is found around 1–5 km altitude between 60–70°N, which coincides with a peak in the L1 backscatter coefficient due to a low altitude layer of smoke. CALIOP weighted extinction heights are found at intermediate heights between these high and low altitude layers.

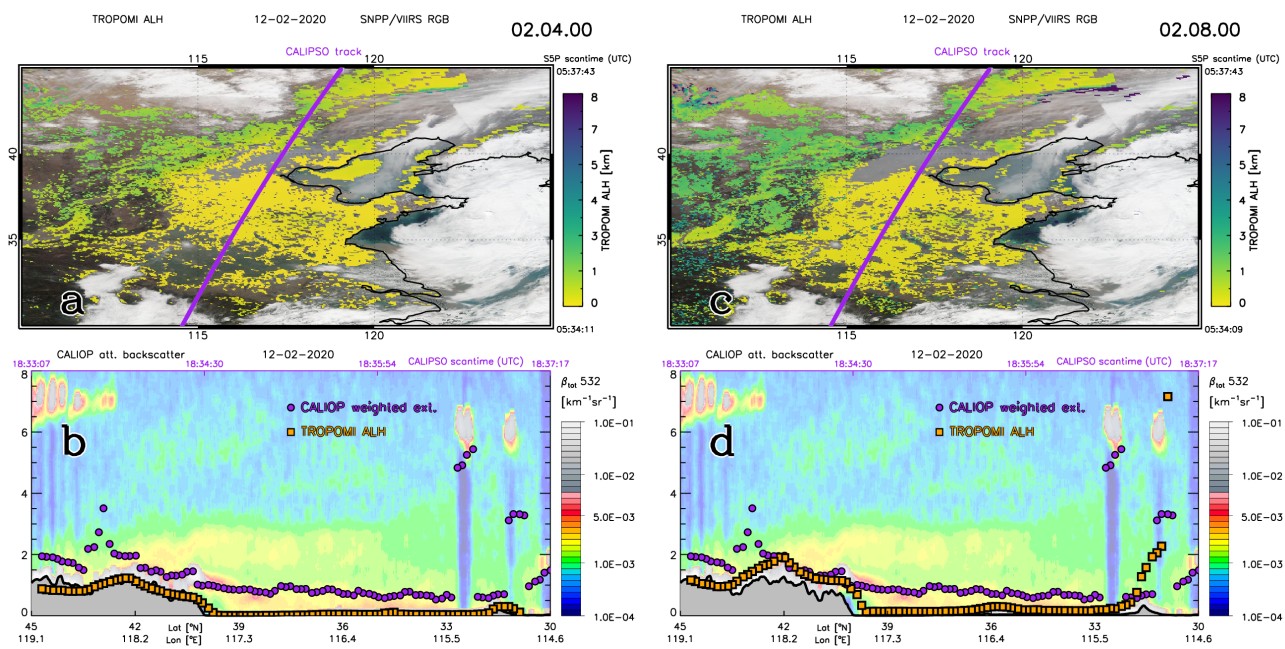

**Figure A5.** Same as Fig A1, but on 12 Feb. 2020 showing industrial pollution over China, from 5:33:48–5:37:48 UTC for TROPOMI data and 18:33:07–18:37:17 UTC for CALIOP data. Note that CALIOP data were collected from the nighttime overpass in order to get a good coverage of the plume over Beijing.

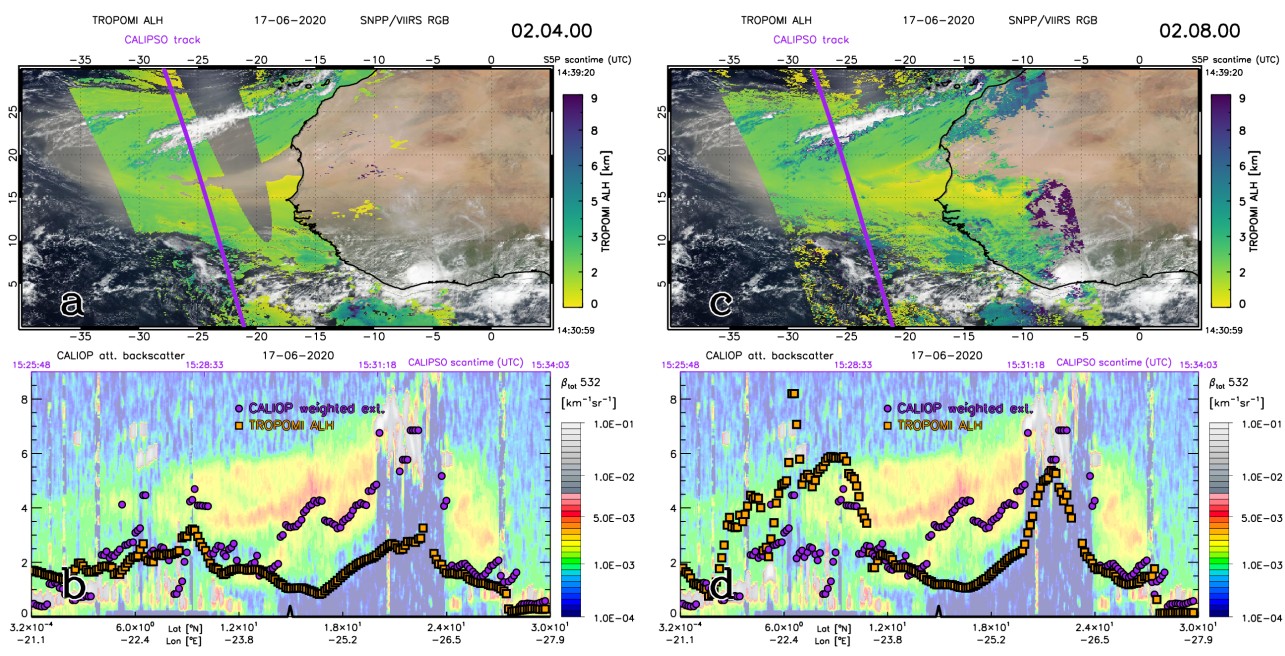

**Figure A6.** Same as Fig A1, but on 17 June 2020 from 14:30:58–14:39:20 UTC for TROPOMI data and 15:25:48–15:34:03 UTC for CALIOP data.

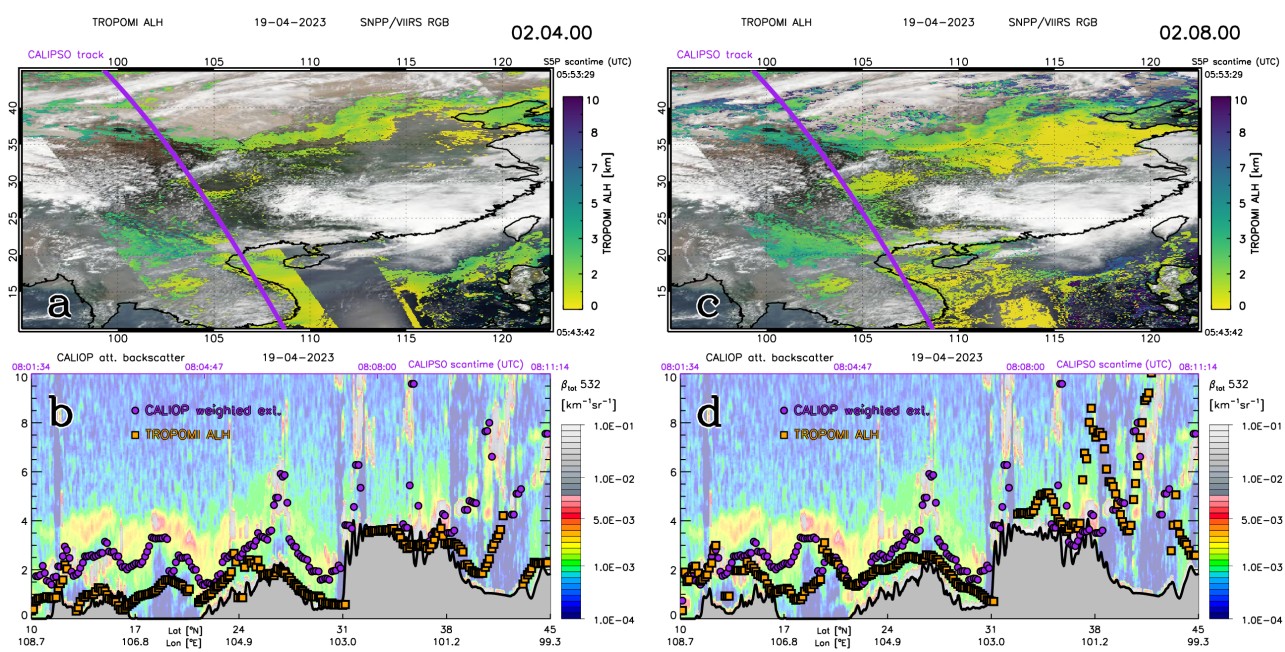

**Figure A7.** Same as Fig A1, but on 19 April 2023 from 5:43:42–5:53:29 UTC for TROPOMI data and 8:01:34–8:11:14 UTC for CALIOP data.

*Data availability.* The Sentinel-5P Level 2 Aerosol Layer Height data are freely available from https://registry.opendata.aws/sentinel5p. CALIOP L1 (NASA/LARC/SD/ASDC, 2024a) and L2 (NASA/LARC/SD/ASDC, 2024b) data are freely available on https://asdc.larc.nasa.gov.

*Author contributions.* MdG wrote the manuscript and validated the data, MS and MtL wrote the operational ALH processor, LGT created
the DLER climatology, DPD developed the EBD data processor from ATLID, GJvZ is PI for the EarthCARE DISC, JPV is TROPOMI PI and developer of the ALH algorithm

*Competing interests.* The authors declare no competing interests

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
