# Peer review of "Improvements in aerosol layer height retrievals from TROPOMI oxygen A-band measurements by surface albedo fitting in optimal estimation"

_Atmospheric Measurement Techniques, 2024_

## Author Comment (AC1)

**Review of "amt-2024-198" by Martin de Graaf, Maarten Sneep, Mark ter Linden, L. Gijsbert Tilstra, and J. Pepijn Veefkind**

Comment on amt-2024-198', Anonymous Referee #1, 29 Jan 2025

Overall this is a significantly well-written article, clear and logical in its premises and in the material presented. The topic described is perfectly adherent to the purpose of AMT as it describes the improvements achieved in deriving the height of aerosol layers by including the surface in the inversion of TROPOMI measurements of the oxygen band.

The reviewer is thanked for the throrough review and critical questions with the intention to improve the paper. This is highly appreciated. All questions and specific comments are addressed below point by point. Where appropriate, the changes in the paper are highlighted and explained.

In the spirit of improving its scientific impact, I list specific comments below. In general, these are minor corrections that should not cause much effort. I would appreciate more contrast and integration with the scientific findings of three relevant papers focusing on retrieval of aerosol layer height. I believe that this would strengthen not only the present paper but also help the community of interested developers and users to advisely both use the product and implement next endeavours. The three papers are Sanders et al (2013, 2015) and Kylling et al (2018).

- Sanders, A. F. J. and de Haan, J. F.: Retrieval of aerosol parameters from the oxygen A band in the presence of chlorophyll fluorescence, Atmos. Meas. Tech., 6, 2725–2740, https://doi.org/10.5194/amt-6-2725-2013, 2013.

- Sanders, A. F. J., de Haan, J. F., Sneep, M., Apituley, A., Stammes, P., Vieitez, M. O., Tilstra, L. G., Tuinder, O. N. E., Koning, C. E., and Veefkind, J. P.: Evaluation of the operational Aerosol Layer Height retrieval algorithm for Sentinel-5 Precursor: application to O2 A band observations from GOME-2A, Atmos. Meas. Tech., 8, 4947–4977, https://doi.org/10.5194/amt-8-4947-2015, 2015

- Kylling, A., Vandenbussche, S., Capelle, V., Cuesta, J., Klüser, L., Lelli, L., Popp, T., Stebel, K., and Veefkind, P.: Comparison of dust-layer heights from active and passive satellite sensors, Atmos. Meas. Tech., 11, 2911–2936, doi:10.5194/amt-11-2911-2018, 2018.

The Kylling paper was missing, which was indeed an omission and is now added as requested. The Sander et al (2013) paper deals with fluorecence which is not considered in the TROPOMI ALH and therefore considered out of scope. Sanders et al (2015) was already present in the manuscript.

The exact questions are:

1. From the contrast with the findings of Sanders et al., can the authors answer whether the surface shall eventually be always fitted?

As Sanders et al (2015) report, fitting the surface albedo increases the convergence, it results in layers higher in the atmosphere and for simulated cases the results were highly improved, in line with our own findings here. The reported contrast the referee refers to is the effect of surface albedo fitting in GOME-2A retrievals (described in Sanders et al (2015)). Kylling et al (2018) compared the results from GOME-2A with ALH from various instruments and found that GOME-2A retrievals had ALH over land that were higher than over the ocean, which is remarkable and

different from all other retrievals. In that case, inclusion of the surface albedo fit increased the height of the layer even further, resulting in a worse comparison with CALIOP. Therefore, it was concluded that fitting the surface albedo was not recommended. However, it seems this conclusion is specific for the application to the GOME-2A retrievals in Sander et al (2015).

About the question whether surface albedo should eventually always be fitted: As explained in Sanders et al (2015), and referenced in the introduction of the current paper, the application of this algorithm with surface albedo fit is likely only successful if an instrument is capable of resolving the O2-A band spectrally and distinguish the derivatives with respect to surface albedo, which drives the OE search. Then results will be less biased to the surface over (bright) surfaces. This is in line with our own experience that the application of the algorithm to S3 OLCI measurements (with only three measurements in the O2-A band) were not improved by adding the surface albedo to the feature vector. This should likely be determined by trial and error for instruments that do resolve the O2-A band spectrally. So the final answer would be: the possibility of fitting the surface albedo should be included, but it must be tested on real instrument data to see whether it does what it should. In this way, we have prepared the algorithm for S4 and S5.

2. From the contrast with the findings of Kylling et al (2018) can the authors make an effort and compare their knowledge of the problem at hand with the discussion points provided in Section 4 of that paper? There, other algorithms based on the fit of the oxygen A band have been compared and some open points have been left unanswered.

The Kylling paper lists a number of questions. What we can learn from the TROPOMI study presented here is that the algorithm is sensitive to instrument characteristics, as discussed above. Also, the TROPOMI ALH product is maturing is such a way that their question "1c: Could an optimal aerosol height algorithm covering all situations be developed?" could be tentatively positively answered. The current implementation of the algorithm for TROPOMI shows retrievals within the retrieval requirement of ± 1 km for almost all situations, as far as we have assessed. The definite answer to this, and to "1b: How will a larger data set in time and space affect the results?" is something that we hope to answer after reprocessing of the TROPOMI data set, but this is not foreseen until 2026. Then, question "1a: How will the results change when including other types of aerosol in the analysis?" is something both reviewers raised, and will be addressed elsewhere in their reviews (below at **P5 L132**).

**Specific comments**

**P2 L53:** here it seems reasonable to add the Kylling et al paper (https://doi.org/10.5194/amt-11-2911-2018).
Agreed

**P3 L79:** it will be interesting to understand if and why the conclusions by Sanders et al 2015 are confirmed or not by this paper.
In line with the Sanders papers, the conclusion is that application of this algorithm without surface albedo fit results in retrievals biased to the surface over (bright) surfaces. Whether this can be solved by including the surface albedo fit should likely be determined by trial and error for

instruments that do resolve the O2-A band spectrally. In case of TROPOMI the surface albedo fitting improved the retrieval over land, but as shown in this paper it still needs some careful expert consideration and fine-tuning, like including a noise floor for TROPOMI and the tuning over ocean surfaces.

**P4 L115:** please add reference to the Kylling er at (2018) paper for consistency. There also other algorithms are described using complementary techniques.
The reference to Kylling was added in the introduction, where other algorithms and techniques are discussed. Here the reader is informed about TROPOMI ALH products not developed at KNMI, which may not share the same characteristics and therefore, the conclusions from the present paper do not necessarily apply to those algorithms and products.

**P4 Section 2.1:** I am not sure that the title of this section is fully appropriate. The first paragraph presents the most important specifications of the sensor, but the second presents the details of the algorithm. I suggest a more descriptive and appropriate title, if the authors do not want to add a separate section.
Agreed. The section was rearranged to first describe the intrument "TROPOMI" and then the algorithm "Aerosol Layer Height product".

About the algorithm and its details: there is an omission in the description and that is the spectroscopy used and especially what form of line (e.g. Voigt, Gaussian, Rautian, speed-dependent and so on) was used to recreate the oxygen band. I invite the authors to provide details. I ask this because depending on the height of the tropospheric column where we place ourselves, one line shape will be more appropriate than others. I am well aware that this is beyond the scope of this paper, but for future reference it is interesting to know how the authors set up the RT calculations.

The reviewer here refers to the forward model, which is computed using the RTM DISAMAR. The RTM is described in De Haan et al (2022), which reads in section 3.2: "The standard database for line-absorbing molecules is the HITRAN 2008 database (Rothman et al., 2009). Line parameters are read from the HITRAN database for a particular gas and a Voigt profile is used to calculate the absorption cross section. For H2O, CO2, CO, and CH4, line mixing is ignored. For O2, line mixing is taken into account using the model described in Tran and Hartmann (2008) for the O2 A-band (De Haan, 2012)"
A sentence similar to the information above, describing the line absorption, was added to the manuscript.

**P5 L132:** Can the authors justify the choice of the Henyey-Greenstein function and the value of asymmetry parametr? First, the advantage of a Mie ot T-Matrix over the H-G phase function is that they better describe aerosol particle scattering. Even more important is the interrelation to the size distribution. Size distribution, asymmetry parameter and single scattering albedo determine the backscattering efficiencies of the particles. For clouds and a g=8.44, for instance, a H-G phase function causes a 60% deviation in backscatter when compared to a Mie phase function (Hansen 1969). For aerosols in accumulation mode, the adoption of H-G gives rise to a 12% discrepancy against Mie-based approach. (Marshall et al., 1995) Moreover, assuming that for typical TROPOMI line-of-sights we are in the backward scattering direction, the H-G phase function would clearly understimate the signal (Fig. 1 in Seidel et al, 2010). This result is in line

with the findings of Marshall et al, because the H-G implies the overestimation of the asymmetry parameter therefore understimating the aerosol signal for the same measurement at TOA.

- J. Hansen, "Exact and approximate solutions for multiple scattering by cloudy and hazy planetary atmospheres," J. Atmos. Sci. 26, 478–487 (1969)

- Stephen F. Marshall, David S. Covert, and Robert J. Charlson, "Relationship between asymmetry parameter and hemispheric backscatter ratio: implications for climate forcing by aerosols," Appl. Opt. 34, 6306-6311 (1995)

- Seidel, F. C., Kokhanovsky, A. A., and Schaepman, M. E.: Fast and simple model for atmospheric radiative transfer, Atmos. Meas. Tech., 3, 1129–1141, https://doi.org/10.5194/amt-3-1129-2010, 2010.

The reviewer's point is well taken, and also raised by reviewer #3. The choice for one, simple aerosol model is prompted by the lack of better information. An operational satellite must rely on available data and currently no dynamic information on aerosol type is available. We could use a climatology or model data, but this may introduce unknown biases due to improper choices of aerosol models.

A simple aerosol model introduces errors, which are reflected in the validation. Furthermore, any differences between the modelled and measured reflectance is mostly compensated by changing the optical depth. In the O2-A band the absorption lines are due to absorption by oxygen, not by the aerosols, which is controlled by the height of the scattering layer. Any erorr in the scattering phase function or single scattering albedo has a second order effect on the retrieved height at most.

Therefore, height is the prime objective and the retrieved aerosol optical depth is considered an effective and secondary quantity, not to be used as an AOT as such.

The following was added to the manuscript as a clarification: "This model does not account for different aerosol types, but the ALH was shown to be robust with respect to fixed aerosol model parameters (sanders et al 2015, nanda et al 2019). The main reason is that differences between the modeled and the measured reflectances are mostly absorbed by the AOT, which is primarily controlled by the fit of the spectra in the continuum. Therefore, AOT is considered an effective quantity and not to be used as an AOT measurement. On the other hand, the ALH is optimized in the retrieval and considered the prime retrieval target. Currently, no dynamic information (daily measurements) on aerosol type is available, but this may change with missions like EarthCARE, PACE and Metop-SG A, in which case a better fit with different aerosol models may be considered for operational processing."

**P6 L161:** For the casual reader, please rephrase or shortly explain what "Pre-whitening is applied" mean.
The original paper read: 'Pre-whitening is applied, using $\mathbf{S}_\epsilon$ to scale the elements of the state vector elements in order to increase numerical stability.'

This is now rephrased into: 'The state vector elements are scaled with $\mathbf{S}_\epsilon$ to bring them in a range that increases the numerical stability, an operation called pre-whitening.'

**P6 L176:** "In section 3.3 the effect of different weights in the a priori error covariance matrix is described for retrievals over ocean." Why only over the ocean and not also for pixels over land, since the authors state these are the cases where they see a gain in accuracy (P4 L94)? I find inconsistent to show the weights for ocean pixels, which have lost accuracy in some cases, while not showing those for land pixels, which are the true improvement of this version of the algorithm.

As described, and shown in Table 2, the *a priori* error over land is set to 0.05, the value over ocean is set to 0.01, after some testing. Over land, the value is large enough to allow the large range of surface albedos encountered over land. As Figure 3 shows, the effect flattens out for larger *a priori* error values: it makes no difference to set it larger than 0.05. The opposite of the choice is not to fit at all, which was our first solution for the ocean cases (setting the *a priori* error to 0.002, effectively limiting a fit to the *a priori* value, which is the surface albedo). This was found to not improve the situation, because the retrievals became noisy, so a more optimal choice was found by trial and error, resulting in fit over oceans with a large contribution of the *a priori* value. Over land, the *a priori* is relaxed so the surface albedo fit can compensate for dark to bright surfaces.

**P6,7 L187-189:** "Note that the ALH does not take different aerosol types into account, but assumes weakly absorbing aerosols, because in the O 2 A-band the penetration depth is controlled by the scattering of the aerosol layer, not the absorption." Fair enough. But again, the scattering part in extinction needs to be correctly assumed. See my comment above about the H-G phase function and size distributions. The variety of aerosol types you analyse cannot be captured by a single aerosol model.
See answer to **P5 L132**. This is now expained more elaborately in the manuscript.

**P7 Eq 4:** Can the authors justify the choice for this definition of aerosol layer height? (see Section 2.1.1 in Kylling et al, 2018)
Kylling et al explain that there is no unique definition of aerosol layer height, even from CALIOP data four different heights are presented in Kylling et al. We chose the extinction weighted height because it was used in Nanda (2020) before and therefore our results can be compared directly with their results. This explanation was added to the manuscript. Griffin et al (2020) compared with CALIOP geometrical heights and results are similar, i.e. TROPOMI is a centroidal height and different from the geometric height.

**P13 L300:** In fact, Figure 4 is even more convincing than Figure 3 in demonstrating the algorithm's improvements. The authors should also not overlook the fact that CALIOP itself is not free from errors arising from the assumption of a lidar ratio that, especially in cases of high optical thickness, does not describe properly multiple scattering (see e.g. Cuesta et al., 2009, 2015). This consideration naturally leads me to manage an expectation of my own, which can be summarized in the following question: How does the algorithm behave as a function of the optical thickness of the aerosol layer? I would like the authors to develop the issue and answer the question above and also this one: how does the addition of the surface in the state vector correlate with the accuracy of the aerosol layer height via the surface-AOT correlation in the oxygen band continuum (i.e., for wavelengths shorter than 758/9 nm)? I suspect that what is gained in fitting the surface is lost in determining AOT, which parameter becomes a de facto error sink. For the avoidance of doubt: I am not asking to validate the AOT derived from the oxygen band. I am clear about its limitations and that a multi-spectral approach is more appropriate. What I am asking is to inspect the trends

between TROPOMI AOT_O2, ALH accuracy (TROPOMI - CALIOP) and goodness of the surface fit. al (2013, 2015) and Kylling et al (2018).

- Cuesta, J., Marsham, J. H., Parker, D. J., and Flamant, C.: Dynamical mechanisms controlling the vertical redistribution of dust and the thermodynamic structure of the West Saharan atmospheric boundary layer during summer, Atmos. Sci. Lett., 10, 34–42, https://doi.org/10.1002/a 2009

- Cuesta, J., Eremenko, M., Flamant, C., Dufour, G., Laurent, B., Bergametti, G., Höpfner, M., Orphal, J., and Zhou, D.: Three-dimensional distribution of a major desert dust outbreak over East Asia in March 2008 derived from IASI satellite observations, J. Geophys. Res.-Atmos., 120, 7099–7127, https://doi.org/10.1002/2014JD022406, 2015

The reviewer is correct in this observation. CALIOP height is not uniquely defined (as outlined in Kylling et al (2028) and recently in Kim et al (2025)), and differences between different CALIOP heights and TROPOMI ALH have been explored in Griffin et al (2020). They investigated TROPOMI ALH as a function of aerosol layer thickness (both optical thickness and geomtric thickness), showing that the difference between CALIOP (geometric and weighted extinction) height and TROPOMI ALH decreases for increasing thickness, as expected. These papers are referenced in the introduction.

The assumption of the AOT as a de facto error sink is correct, as outlined in previous answers.

**Typos and style**

**P6 L161:** "to scale the elements of the state vector elements". Perhaps a repetition? Indeed, corrected

**P6 L181:** "a set of nice different cases". Perhaps nine? Indeed, corrected

**P6 L186:** "CALIOP L1 data". Spurious sentence. Indeed, removed

---

## Author Comment (AC2)

**Comment on "amt-2024-198" by Martin de Graaf, Maarten Sneep, Mark ter Linden, L. Gijsbert Tilstra, and J. Pepijn Veefkind**

RC2: 'Comment on amt-2024-198', Anonymous Referee #3, 12 Mar 2025

The authors present a new aerosol layer height (ALH) product from TROPOMI oxygen A-band measurements. This product employs surface albedo estimated from TROPOMI measurements (as opposed to previous versions that used GOME-based albedo data). This is an important change as it enables usage of the correct viewing geometry (TROPOMI makes measurements in the afternoon while GOME does so in the morning) and hence provides the proper directional reflectivity. Comparisons with CALIPSO measurements demonstrate significantly improved ALH retrievals over land (including bright surfaces) and decreased land-ocean contrast. This work is novel and well written and certainly publishable in AMT.

The reviewer is thanked for kind review and assesment. Below we address all the points raised and answer the questions. The changes made in the manuscript are highlighted.

**I have only two major comments.**

First, it is mentioned that the neural network training was done assuming fixed aerosol properties, in particular a single scattering albedo of 0.95 and a Henyey-Greenstein function with asymmetry parameter 0.7. Does this not bias the retrievals when the actual aerosols present have different properties? The authors note that "the ALH does not take different aerosol types into account, but assumes weakly absorbing aerosols, because in the O2 A-band the penetration depth is controlled by the scattering of the aerosol layer, not the absorption." I am not so sure that this is true (and even if it is, the asymmetry parameter would matter). The single scattering albedo and the phase function do affect the relative interplay between aerosol scattering and gaseous absorption. At the very least, the authors should do some sensitivity studies (varying SSA and asymmetry parameter) to prove their hypothesis that the absorption does not matter.

This is a valid point, also raised by reviewer #1. The explanation in the manuscript was too brief and neglected the considerations leading to the argument. This has been addressed in the manuscrpt, explaining the sensitivity. In short, differences between the modelled and meaured reflectances, such as introduced by differences in the real and modeled aerosol model and scattering phase function, are compensated by changing the aerosol optical thickness, from fitting the reflectances in the continuum. Therefore, the AOT from the retrieval is considered an effective AOT and not to be used as an AOT measurement. See also the reply to Reviewer #1.

The following was added to the manuscript as a clarification: "This model does not account for different aerosol types, but the ALH was shown to be robust with respect to fixed aerosol model parameters (sanders et al 2015, nanda et al 2019). The main reason is that differences between the modeled and the measured reflectances are mostly absorbed by the AOT, which is primarily controlled by the fit of the spectra in the continuum. Therefore, AOT is considered an effective quantity and not to be used as an AOT measurement. On the other hand, the ALH is optimized in the retrieval and considered the prime retrieval target. Currently, no dynamic information (daily measurements) on aerosol type is available, but this may change with missions like EarthCARE, PACE and Metop-SG A, in which case a better fit with different aerosol models may be considered for operational processing."

Second, some of the aerosol plume events have an hour or longer time difference between the Sentinel-5p and CALIPSO overpasses. What was the purpose of selecting these cases? There needs to be some text describing the rationale for the case selection.

The selection of the cases is based on several criteria, involving temporal and spatial coverage, (globally and during the entire TROPOMI mission as much as possible), different aerosol events and coverage over land and ocean surfaces. CALIPSO and Sentinel-5P have similar equator crossing times, resulting mosly in small time differences between CALIOP and TROPOMI measurements (depending on the details of the CALIPSO track in the swath of TROPOMI, which is 2600 km wide). Since we follow the validation excercise by Nanda et al (2020), time differences up to 5 hours are acceptable, also considering that aerosol plumes are generally not very dynamic and can be expected to mainly move laterally to first order. This is easily satisfied by all cases but one, (over Asia in 2020), which turned out to have a very poor coverage of CALIOP measurements during daytoime, and we decided to show the more interesting coimparison with the night time overpass. This was mentioned in the caption of Figure A5: "Note that CALIOP data were collected from the nighttime overpass in order to get a good coverage of the plume over Beijing."
For reference, the coverage with the daytime overpass is shown here in Fig. 1, to illustrate the poor selection (Calipso track is just in the low-left corner). Furthermore, it also shows that the

[Figure]

Figure 1: (a) True color (RGB) image from SNPP/VIIRS on 12 Feb. 2020 showing industrial pollution over China,, overlaid with TROPOMI ALH, version 02.04.00 from 14:37:59–14:43:35 UTC. The purple line shows the daytime CALIPSO track over the area on the same day from 06:11:51–06:13:10 UTC. (b) CALIOP L1 532 nm attenuated backscatter curtain image along the purple track in the top panels, overlaid with the CALIOP weighted extinction height (purple dots) from L2 extinction profiles at 532 nm (averaged every 0.15° latitude along the track) and the average TROPOMI ALH of collocated pixels within a 0.5° radius of the CALIOP extinction profiles along the track. (c) and (d): Same as (a) and (b), but with TROPOMI ALH version 02.08.00

comparison does not necessarily show better or worse results, so we feel it justifies to keep the figure A5 as it is, also to show comparisons are similar even after 13 hours.
The explanation of the selection of cases was extended in the manuscript (section 2.4).

**Minor comments/typos:**

**Line 37:**  space-based instruments → space-based retrievals
Indeed, corrected

**Line 42:**  extend → extent
Corrected

**Line 54:**  remove "like"
Agreed

**Line 65; Line 104:**  wavelengths → wavelength
Both corrected

**Line 114:**  20204 → 2024
Corrected

**Line 123:**  For the ALH → For the ALH retrieval,
Agreed

**Line 174:**  maximum likelihood → maximum likelihood estimate
Agreed

**Line 185:**  remove "CALIOP L1 data"
Corrected

**Lines 212-213:**  land surfaces and ocean surfaces → land and ocean surfaces
Corrected

**Line 303:**  weakly scatterers → weak scatterers
Corrected